# Clinical and Humanistic Outcomes of Community Pharmacy-Based Healthcare Interventions Regarding Medication Use in Older Adults: A Systematic Review and Meta-Analysis

**DOI:** 10.3390/healthcare9111577

**Published:** 2021-11-18

**Authors:** Christina Malini Christopher, Bhuvan KC, Ali Blebil, Deepa Alex, Mohamed Izham Mohamed Ibrahim, Norhasimah Ismail, Alian A. Alrasheedy

**Affiliations:** 1School of Pharmacy, Monash University Malaysia, Subang Jaya 47500, Malaysia; christina.christopher@monash.edu (C.M.C.); aliblebil@yahoo.com (A.B.); 2Jeffrey Cheah School of Medicine and Health Sciences, Monash University Malaysia, Subang Jaya 47500, Malaysia; deepa.alex@monash.edu; 3Clinical Pharmacy and Practice Department, College of Pharmacy, QU Health Qatar University, Doha 2713, Qatar; mohamedizham@qu.edu.qa; 4Bayan Lepas Health Clinic, Bayan Lepas 11900, Malaysia; asmak4468@gmail.com; 5Department of Pharmacy Practice, College of Pharmacy, Qassim University, Buraidah 51452, Saudi Arabia

**Keywords:** community pharmacy, intervention, older adults, outcomes, systematic review

## Abstract

This review and meta-analysis aimed to determine the clinical and humanistic outcomes of community pharmacy-based interventions on medication-related problems of older adults at the primary care level. We identified randomized controlled trials (RCTs) examining the impact of various community pharmacy-based interventions from five electronic databases (namely, MEDLINE (Ovid), EMBASE (Ovid), CINAHL, APA PSYInfo, and Scopus) from January 2010 to December 2020. Consequently, we assessed these interventions’ clinical and humanistic outcomes on older adults and compared them with non-intervention. We included 13 RCTs in the current review and completed a meta-analysis with six of them. The included studies had a total of 6173 older adults. Quantitative analysis showed that patient education was significantly associated with an increase in the discontinuation of sedative–hypnotics use (risk ratio 1.28; 95% CI (1.20, 1.36) I2 = 0%, *p* < 0.00001). Moreover, the qualitative analysis showed that medication reviews and education with follow-ups could improve various clinical outcomes, including reducing adverse drug events, reducing uncontrolled health outcomes, and improving appropriate medication use among the elderly population. However, medication review could not significantly reduce the number of older adults who fall (risk ratio 1.25; 95% CI (0.78, 1.99) I2 = 0%, *p* = 0.36) and require hospitalization (risk ratio 0.72; 95% CI (0.47, 1.12) I2 = 45%, *p* = 0.15). This study showed that community pharmacy-based interventions could help discontinue inappropriate prescription medications among older adults and could improve several clinical and humanistic outcomes. However, more effective community pharmacy-based interventions should be implemented, and more research is needed to provide further evidence for clinical and humanistic outcomes of such interventions on older adults.

## 1. Introduction

There is an ever-increasing need for healthcare services for older adults because of the increase in the aging population. The population of older adults (65 and above) was estimated to be 8.5% of the total population (i.e., 617.1 million) in 2015 and is expected to reach 12% in 2050 (i.e., 1 billion) [1]. The prevalence of multiple chronic illnesses that require comprehensive and complex care is higher in this population. Accordingly, older adults consume a high proportion of prescription medicines and over-the-counter (OTC) medicines and take multiple medicines to manage their chronic illnesses [2,3,4]. Health-related problems arise when older adults do not take medicines as prescribed, self-consume medicines, or consume the wrong medicines for various reasons [5].

Medication-use problems of older adults are complex and multifaceted and cause an enormous public health, social, and financial burden to the economy [6,7]. Medication usage problems of older adults can affect the optimal therapeutic outcomes and cause adverse drug events and serious harm. The problems related to medication usage in older adults happen at both secondary/tertiary and primary care levels. In the hospital setting, the involvement of multiple healthcare professionals, via a collaborative care model, and the focus on medication safety can help identify and minimize medication-related problems of older adults. In contrast, in primary care settings, the approach of healthcare delivery mostly focuses on preventing illness and promoting health [8]. In general, the primary care level lacks a geriatric-focused care delivery that can identify complex healthcare and medication usage need of older adults and support them adequately.

Medication usage for the older adults at the primary care level is coordinated via general practitioners (both private and government primary health clinics), community nurses, and community pharmacists. Furthermore, the transition of care for older adults happens from secondary and tertiary healthcare to primary healthcare facilities [9]. Consequently, community pharmacies are a pivotal junction in this entire paradigm, responsible for delivering medications and ensuring appropriate use of medications among older adults.

Several studies have examined the problems of medication use of older adults and the potential for community pharmacists to contribute to appropriate medication use at the primary care level. Studies have reported a post-discharge medication review by community pharmacists and its impact on the aging population [10,11,12]. A study by Kayyali et al. [13] in the UK has reported problems among older adults such as difficulty in medication administration (40%), lack of monitoring of patients with diabetes, and risk of falling (14.3%). Another study by Foubert et al. [14] conducted among community-dwelling older adults (patients) with polypharmacy and those receiving home health care with medication schemes’ altercation (review) by community pharmacists showed that pharmacists’ interventions enabled more complete and accurate medication schemes. Several reviews have highlighted the improvement in medication adherence among older adults following an intervention by community pharmacists [15,16,17]. Apart from medication adherence, there was improved quality of life and reduced drug-related problems from these reviews.

Overall, several studies have reported improved health outcomes from various pharmacists’ interventions on older adults’ medication use [18,19]. Some of these interventions were delivered by pharmacists during the transition of care as a collaborative care model with community pharmacists, while some are delivered solely via community pharmacy-based interventions. A systematic review by Cooper et al. [20] regarding pharmacists’ interventions to improve appropriate use of polypharmacy among older adults did not find significant clinical improvements. However, the systematic review evaluated pharmacists’ interventions from both primary and secondary care settings. Likewise, another systematic review by Clyne et al. [21] on pharmacists’ interventions to address potentially inappropriate prescribing in community-dwelling older adults reported that such interventions were beneficial in reducing potentially inappropriate prescribing but with modest effect size. This systematic review included pharmacist’s intervention from different settings, not just the community pharmacy [21]. Thus, from a health system perspective, there is still a need for studies that thoroughly evaluate community pharmacy-based services’ impact with an exclusive focus on older adults’ medication usage problems and relevant clinical and humanistic outcomes. Therefore, this systematic review and meta-analysis aimed to determine the clinical and humanistic outcomes of community pharmacy-based interventions for older adults to solve their medication usage problems. We believe this review will provide evidence for creating and funding a community pharmacy-based appropriate medicine usage support program for older adults.

## 2. Materials and Methods

The study protocol has been registered at PROSPERO 2021 CRD42021229948 and was developed based on the Cochrane Handbook for Systematic Reviews of Interventions and the Preferred Reporting Items for Systematic Reviews and Meta-Analyses (PRISMA) guidelines [22].

### 2.1. Eligibility Criteria

Studies which specifically included population of older adults aged 65 years and above were eligible for the review. Moreover, community pharmacy-based interventions were the main inclusion criteria. Comparator or control was based on non-intervention or not receiving community pharmacy-based services. The outcome was based on interventions regarding medication use among older adults. Study designs of included studies were randomized controlled studies. The exclusion criteria were studies published in a language other than in English and before the year 2010, studies that are not randomized controlled studies, and studies that are not community pharmacy-based interventions.

### 2.2. Search Strategy

The electronic search was performed in MEDLINE (Ovid), Ovid EMBASE, CINAHL, APA PSY Info, and Scopus. The search was for original articles describing community pharmacy-based interventions for older adults regarding medication use from January 2010 to December 2020. (Refer Appendix A). The search process was taken in three steps. The initial search was completed using Scopus and Medline to explore the literature and become more familiar with the terms and current studies—including analyzing each word/term in the titles and abstracts and identifying index terms in each article. After that, in the second step, a comprehensive search was completed by using all index terms and identifying key terms by using the selected databases. In the third step, references of key articles were searched for additional studies. Studies were restricted to the English language. In addition, grey literature was explored to find any potential studies relevant to the study objectives and eligibility criteria. The entire actual search is available in Appendix A.

### 2.3. Study Selection

Two reviewers (C.M.C. and B.K.C.) screened and reviewed the titles and abstracts of identified studies using the search strategy and those from additional sources (i.e., references of retrieved articles, grey literature, and websites from professional pharmacy societies such as Malaysian Pharmaceutical society) to identify studies that meet the inclusion criteria mentioned earlier. Full-text articles were also screened in the same manner. Any disagreements were resolved by consensus through another reviewer (A.B.). Interventions were included if they were community pharmacy-based, and the study design was a randomized controlled trial. Consequently, other studies not meeting these criteria were excluded, including review articles and conference abstracts.

### 2.4. Data Extraction

The first author (C.M.C.) extracted data using a standardized form and was checked by the second author (B.K.C.). Data extracted included publication details (author, year of publication, and journal name); study design characteristics (study design, sample size, objectives, country); study characteristics (type of intervention, method of intervention, and outcome of intervention); and the main results of the study.

### 2.5. Risk of Bias (Quality Assessment)

Two authors (C.M.C and B.K.C.) independently assessed the risk of bias using Cochrane Risk of Bias (ROB 2.0) for randomized controlled trials, which is a revised Cochrane tool [23]. The main domains where bias could arise and judgment of risk of bias needed to be completed include randomization process, deviation from intended interventions, missing outcome data, measurement of outcome, and the selection of reported results. Consequently, based on the risk-of-bias judgment of each domain in the clinical trial, the overall risk of bias can be judged as low risk of bias, some concerns, or high risk of bias. During the judgment of risk of bias, if there were any discrepancies, both reviewers discussed and resolved them. Moreover, we used the GRADE criteria to assess the quality of evidence for each outcome reported [24].

### 2.6. Data Analysis

Studies were eligible for the meta-analysis if at least two outcomes were comparable. Cochrane handbook was used as a guide to analyzing our data [25]. Statistical heterogeneity was assessed using the I^2^ statistic, one of the statistical tools to be present in the meta-analysis study [26]. Heterogeneity was defined as high if I^2^ > 75% and low if I^2^ < 25% [27]. We used a random-effect model in our meta-analysis, assuming that heterogeneity exists within the samples. Results were presented with a risk ratio for the dichotomous variable with a confidence interval of 95%. As a priori, we performed subgroup analyses by the duration of follow-up to review the number of older adults hospitalized. All analyses were performed using Cochrane Review Manager version 5.4. (The Nordic Cochrane Centre, Copenhagen, Denmark).

## 3. Results

A total of 6917 articles were identified through the selected databases. Another nine articles were retrieved from other sources such as Google Scholar for grey literature, manual search in the key references retrieved, and other websites particularly the Ministry of Health Malaysia and the Malaysian Pharmaceutical Society website. After removing duplications (*n* = 1337), a total of 5589 articles were identified for the title and abstract screening, and 108 articles were included for further review by accessing the full texts and assessing them against the inclusion criteria. Most full texts were excluded because of a non-randomized controlled study design (*n* = 49) and non-community pharmacy-based intervention (*n* = 17). Consequently, 13 randomized controlled trials (RCT) were included in this systematic review. Reasons for the exclusion of full texts and the flow of studies are described in Figure 1. In this review, the inter-rater reliability for the final extraction between two reviewers was 0.918.

### 3.1. Characteristics of Included Studies

Among the 13 randomized controlled trials, 7 were cluster randomized control trials, 1 was a double-blind RCT, 1 was a single-blind RCT, 3 were RCTs, and 1 was a pilot RCT, as summarized in Table 1. Trials were carried out in Croatia (two studies), the Netherlands (two studies), the USA (two studies), Canada (two studies), Spain (two studies), New Zealand (one study), Denmark (one study), and Finland (one study) The interventions were conducted by community pharmacists either in a community pharmacy or at patient’s home or medical center clinics or home care unit. The included studies had a total of 6173 older adults with a sample size ranging from 39 to 715 participants. In terms of the type of interventions provided, most of the community pharmacy-based interventions were medication review (*n* = 7), education (*n* = 4), pharmaceutical care (*n* = 1), and electronic device reminder (*n* = 1). In terms of the type of measured outcomes, there were various outcomes reported by studies. Some studies reported the impact on hospitalization (*n* = 4), number of potentially inappropriate medicines (PIM) (*n* = 3), rate of sedative–hypnotics use (*n* = 2), time in warfarin therapeutic range (*n* = 1), quality of life (*n* = 1), medication appropriateness (*n* = 1), drug burden (*n* = 1), rate of discontinuing fall-risk inducing drug (*n* = 1), number of adverse drug events (*n* = 2), mortality (*n* = 1), medication adherence (*n* = 2), and uncontrolled health problems (*n* = 1). The details are presented in Table 1.

### 3.2. Risk of Bias Assessment

Figure 2 shows the risk of bias of all the included studies in each domain. Overall, four studies were judged to have a low risk of bias, seven studies were judged to have some concerns regarding the level of risk, and two studies had a high risk of bias. Five studies raise some concerns regarding potential biases in the randomization process [28,29,30,31,32]. All studies did not deviate from intended interventions. Only one study reported some concerns regarding biases on missing outcome data [33]. Two studies did not report a measurement of outcome [32,34]. Some studies were judged to have a selection of reporting biases but were judged to have “some concerns” [28,29,30,31,32,33,35,36].

### 3.3. Types of Community Pharmacists’ Interventions

#### 3.3.1. Medication Review

Most of the studies (*n* = 7) performed medication reviews as their main intervention. Comprehensive medication reviews were initiated by interviewing older adults; screening their medication list, lab values, and complementary medicines; and a pharmacotherapeutic plan was decided [28,30,31,32,34,35,37]. Then, the plan was discussed with prescribers and patients. Finally, the plan was executed with follow-up monitoring by the community pharmacist [28,30,31,32,34,35,37]. One study had implemented medication review as their main intervention under the coordination of care with other primary health care providers [34]. The study by Touchette et al. [31] included two groups as interventions: basic medication review care and medication review enhanced care. The difference between both is that the latter group had access to clinical information regarding laboratory values of patients, and the former did not have access to the information. In this review, we included the medication review enhanced care as the intervention since it included a comprehensive review with patient lab findings and usual care as a control group.

#### 3.3.2. Educational Intervention

Four RCTs examined the impact of educational intervention [29,36,38,39]. Under this intervention, two studies included a follow-up plan. Participants were provided with a form containing lab values, INR, and important education points and were given a pillbox [29,39]. Tannenbaum et al. [36] provided patient education materials that also contained a tapering benzodiazepine dose in a separate study. Martin et al. [38] mentioned that their study included education materials, including on tapering benzodiazepine dose, distributed to patients and prescribers provided with basic educational materials.

#### 3.3.3. Pharmaceutical Care

This intervention was undertaken by a community pharmacist initially examining the medication list of older adults, answering any questions on their medications, and providing leaflets and motivational adherence support [33]. Older adults would then be followed up after 3, 6, and 9 months, and any drug-related problems involved consultation with prescribers. This intervention differs from a comprehensive medication review because it includes no pharmacotherapeutic plan to be discussed with the prescribers before dispensing the medications to older adults.

#### 3.3.4. Electronic Reminder Device

Only one study implemented this intervention with brief counseling to assess whether it improved refill adherence and persistence for statin treatment in non-adherent older adults [40].

### 3.4. The Outcomes of the Interventions

#### 3.4.1. Hospitalization

Three studies specifically examined the impact of medication review on the hospitalization of older adults [31,35,37]. In one of the studies, Touchette et al. [31] reported outcomes based on a shorter follow-up duration of three months and a longer duration of six months. The quantitative analysis showed that these three pooled studies did not show a statistically significant impact of medication review on the probability of hospitalization (risk ratio 0.72; 95% confidence interval (0.47, 1.12) I^2^ = 45%, *p* = 0.15) (Figure 3).

#### 3.4.2. Sedative–Hypnotics Users

Two studies were using sedative–hypnotics (benzodiazepines) as their outcome assessment drug [36,38]. Both studies were pooled, and patient education was statistically significant for reducing the number of sedative–hypnotic users (risk ratio 1.28; 95% confidence interval (1.20, 1.36) I^2^ = 0%, *p* < 0.00001). (Figure 4).

#### 3.4.3. Number of Older Adults Who Fall

Two studies were pooled to assess the medication review intervention on the number of older adults’ falls [30,37]. Both studies were not statistically significant for reducing the number of older adults who fall (risk ratio 1.25; 95% confidence interval (0.78, 1.99) I^2^ = 0%, *p* = 0.36) (Figure 5).

#### 3.4.4. Potentially Inappropriate Medications

Three studies reported the number of potentially inappropriate medicines as the outcome of the intervention. Martin et al. [38] reported that at 6 months, 43% in the intervention group did not have prescriptions for inappropriate medicines compared with only 12% in the control group. Moreover, Bryant et al. [28] reported that the mean number of inappropriate medicines per patient was higher for the intervention group at baseline (2.5) and reduced after 6 months of intervention (2.5 versus 1.6, respectively, *p* < 0.001) compared to the control group (2.1 versus 2.1, respectively, *p* = 0.991). Another study by Toivo et al. [34] did not have significant findings based on their intervention on the potentially inappropriate medication. However, the role of pharmacists in this study was part of coordinated care involving other healthcare professionals. The results are summarized in Table 1.

#### 3.4.5. Medication Adherence

One RCT that examined the impact of one type of pharmaceutical care did not report a significant impact on medication adherence as per the study by Olesen et al. [33]. Similarly, with the electronic reminder device intervention, no improvement of refill adherence was found in the older adults’’ population [40].

#### 3.4.6. Adverse Drug Events

Two studies measured the impact of the interventions in terms of adverse drug events. One study by Falamic et al. [29] highlighted that adverse drug reactions were significantly lower in the group of older adults who were prescribed warfarin and were receiving an educational intervention. The author described that providing patient education on warfarin, pillbox, and a follow-up plan reduced the risk of bleeding as an adverse drug event. Meanwhile, Touchette et al. [31] mentioned no significant impact on adverse drug events after providing medication reviews. However, overall, community-pharmacy-based interventions managed to reduce the number of adverse drug events through patient education.

#### 3.4.7. Other Outcomes

A study by Bryant et al. [28] stated that through medication review, the medication appropriateness index improved, but it did not improve the quality of life in the intervention group. Meanwhile, Mott, Martin [30] described that the intervention group had a significant impact by leading to a higher rate of discontinuing fall-risk-inducing drugs among older adults after medication review completed by the community pharmacist. Another study by Varas–Doval et al. [32] had a significant reduction in the number of uncontrolled health problems after the same intervention. Despite that, Olesen et al. [33] reported no significant improvement in mortality rate after medication adherence was completed. As for educational interventions, Falamic et al. [39] pointed out that it improved warfarin’s therapeutic time range in older adults. As a whole, various community pharmacy-based interventions show improvement in clinical outcomes among older adults. However, evidence is lacking regarding patient satisfaction and quality of life in these studies.

### 3.5. Sensitivity Analysis

Moderate heterogeneity, I^2^ = 45%, was found in the outcome of hospitalization. Thus, a one-on-one removal of studies in the meta-analysis was completed by removing a study by Touchette et al. [31] in the hospitalization outcome, and subsequently, no heterogeneity was found. This analysis reported that medication review was significant for reducing hospitalization of older adults (risk ratio 0.59; 95% confidence interval (0.39, 0.88) I^2^ = 0%, *p* = 0.01. (Figure 6)

### 3.6. Subgroup Analysis

Subgroup analysis comparing studies that reported hospitalization after three months and six months medication reviews were completed (Table 2). The effect of the intervention was not statistically significant between the duration of follow-up of subgroups (risk ratio 0.74; 95% confidence interval (0.54, 1.00) I^2^ = 18%, *p* = 0.05 (Figure 7).

### 3.7. Certainty of Evidence

Based on GRADE criteria, the certainty of the evidence was rated as moderate for the outcome of hospitalization. Outcomes of the number of older adults who fell and ceased benzodiazepine were rated as high-quality based on GRADE criteria.

## 4. Discussion

To the best of our knowledge, this meta-analysis is probably the first study that focused exclusively on the impact of community pharmacy-based interventions on medication use and related clinical and humanistic outcomes among the older population (i.e., 65 years and over). Previous studies have focused on a particular intervention that was carried out within and outside community pharmacy settings and focused on medications usage problems of both the general population and older adults [41,42,43,44]. This review focused on various community pharmacy-based services/interventions for medications usage problems of older adults and its impact based on the best available evidence (i.e., RCTs).

Previous reviews (systematic review and meta-analysis) have focused on various interventions by pharmacists regarding medication usage problems. However, they looked at the outcomes of pharmacists’ interventions regarding medication usage problems of both the general population and older adults (i.e., mixed populations) and reviewed interventions that were carried out by pharmacists working in both primary and secondary care settings (i.e., different settings) [45,46,47,48]. These reviews focused on all types of community pharmacy-based services, and these were not exclusively focused on any specific sub-group of the population. Thus, so far, only one systematic review by Tasai et al. [16] focused on the medication usage problems of the elderly population; however, it also only looked at the impact of medication review on one service (i.e., polypharmacy) in the elderly population. The current systematic review and meta-analysis are different from other studies and reviews as they critically review all the eligible community pharmacy-based services, focusing on medication usage problems of older adults in particular.

Community pharmacy-based interventions were regarded as one of the most accessible primary services by the older population. Most of the interventions were provided by community pharmacies in collaboration with other healthcare professionals, including physicians, general practitioners, and nurses. As a whole, there is evidence in the literature that community pharmacy-based interventions impacted several clinical outcomes among older adults, including reducing inappropriate medicine use (including that of sedative–hypnotics); reducing uncontrolled health problems; and potentially reducing ADRs. However, evidence was lacking in terms of the impact on patient satisfaction and quality of life. Consequently, given the limited literature for studies focused on the elderly population in the community pharmacy setting (i.e., RCTs), more research is needed in this area.

The current review showed that patient education delivered by community pharmacists increased the number of older adults who benefited from the pharmacists’ interventions and discontinued their sedative–hypnotic drugs. For example, we found that patient education improved the cessation of benzodiazepine among older adults. This is in line with a previous review in which education improved the number of older adults who ceased benzodiazepine [49]. However, Reeve, Ong [50], in their review, reported that the rate of benzodiazepine discontinuation was lower with patient education compared with other interventions. However, the differences could be explained by the fact that patient education could be provided in different ways, and hence its impact could be different depending on the type, structure, and nature of the educational intervention. We have noted that the patient education included in our review was thorough and innovative. For example, it was provided together with the visual tapering dose, which is a method of effective intervention leading to the reduced use of benzodiazepine. Brochures on educational materials have also influenced the choices of hypnotic–sedative users. In addition, patient education helps provide knowledge on sedative–hypnotic medication, including its risks and side effects for patients. Consequently, this information provides evidence for supporting this professional service and could be further expanded as part of a collaborative health care model in the primary care setting.

Various community pharmacy-based interventions were identified in this systematic review and meta-analysis. Among various interventions, medication review was the most common intervention carried out by the community pharmacists and was evaluated in RCTs. Varas–Doval et al. [32] reported that medication review with follow-up resulted in a significant reduction in the number of uncontrolled health problems over 6 months in the intervention group compared to no change in the control group. Moreover, it was shown that medication reviews and education by community pharmacists targeting elderly patients resulted in better outcomes in terms of appropriate medication use [28,38]. However, medication reviews by community pharmacists did not reduce the probability of hospitalization among older adults, in contrast with the previous findings of Tasai, Kumpat et al. [16] and Jokanovic et al. [17] on this outcome. However, our findings were on par with several other reviews [51,52,53]. This is possibly because there are only limited studies with moderate heterogeneity in the literature, which cause a non-significance impact; however, significant findings were noted when sensitivity analysis was completed in our review. In the literature, similar to the hospitalization outcomes, mixed results were reported regarding the impact of medication reviews on the risk of falls. Our current study showed that medication reviews did not reduce the number of older adults falling. Similar to our findings, Hart, Phelan [54], and colleagues pointed out that medication review did not reduce the number of older adults who fall in their review. However, another study by Huiskes et al. [51] indicated that medication review decreased the number of older adults falling. Several factors could explain the differences among the results, including self-reporting. The two studies included in our meta-analysis were based on patient self-reporting findings [30,37]. Thus, the possibility of not accurately revealing the number of older adults who fall is high due to old-aged patients’ frail conditions. Moreover, the study by Mott et al. [30] was a pilot study, and the sample size was limited. Thus, this might explain the non-significance results. Consequently, we believe more research is needed to further investigate the impact of community pharmacists’ intervention on these outcomes. In addition, more structured and tailored interventions are needed to be established at the community pharmacies to provide quality services to the elderly population.

There was evidence of a reduction in the inappropriate medications in one of the included studies through patient education on other outcomes [38]. Moreover, Falamic et al. [29] provided education with a pillbox, and adverse drug events were significantly reduced among older adults prescribed warfarin. Kallio et al. [15] justified that most studies showed improvement in medication adherence and reduced drug-related problems among older patients after medication review intervention is completed by community pharmacists. Our systematic review shows some evidence of improvement in medication adherence because of community pharmacy-based interventions. However, when it comes to long term impact, such as the effect of community-pharmacy-based interventions on quality of life, we found limited evidence, with one study by Bryant et al. [28] not showing an impact, which is similar to the results from Huiskes et al. [51]. Several recent studies have investigated telephone calls and automated telephonic prompts as a digital tool, and medication adherence was the most included topic in the digital conversation between community pharmacists and patients [55]. However, these studies did not include any specific population, and there were no studies found on social media platforms as a digital tool, especially involving the older adult population.

Overall, our findings revealed that community pharmacy services are beneficial to older adults to optimize proper medication use, reduce unnecessary benzodiazepine use, reduce uncontrolled health problems, and ADRs among older adults. Therefore, evidence-based educational interventions should be encouraged in community pharmacies to achieve rational medication use among older adults visiting community pharmacies and to provide further improvements in their health.

### 4.1. Strengths and Limitation

The current review has some points of strength. First, we have only considered randomized, controlled trials in our review, and this increased the robustness of the study [56]. Furthermore, we have assessed the outcomes of quantitative analysis through GRADE criteria and only included moderate- and high-quality studies. Heterogeneity across studies was also assessed with sensitivity analysis, and we have reported the homogeneity of studies after removing one study. Our search narrowed to community pharmacy-based interventions, focusing on the older population, which was an added advantage to review older adults’ health care outcomes. However, there are several limitations to this review. Firstly, the search articles for this review were restricted to the English language. Thus, we acknowledge that there might be a limitation to the search in non-English native regions. Secondly, the age limit for older adults in this review was 65 years and above. Therefore, we could not have captured studies that included older adults in the range of 60 years and above. Thirdly, our review resulted in various interventions with different features in terms of the duration, nature, and components of the intervention. Lastly, we could not determine the pooled estimates for other outcomes—medication adherence, quality of life, potentially inappropriate medication, and adverse drug events—because of different outcomes with various interventions. In addition, there were limited RCTs on several outcomes of interest such as adherence, quality of life, etc. However, overall, we believe the current review and meta-analysis provided useful data for future guidance to improve the pharmaceutical care services provided to the older population at community pharmacies.

### 4.2. Implications for Research and Practice

It is well known that community pharmacy is one of the most accessible health care resources and could play a fundamental role in the health care of older adults in a community [16]. However, there are limited RCTs that evaluated the impact of interventions and services in community pharmacies on the health outcomes of populations aged 65 years and over. Consequently, given the rapid surge in the aging population, more future research is needed to implement pharmacists’ interventions and evaluate their clinical, humanistic, and economic outcomes among older adults. In addition, more qualitative exploration focusing on older adults’ mobility, hearing, etc., and other access problems at community pharmacies should be explored. The role of a community pharmacy in certain lower- and middle-income countries (LMICs) still lacks the recognition as primary health care providers. Many older adults in these regions still access tertiary health care as their first point of care. Therefore, this review will help provide an evidence for the development of community pharmacy-based interventions focused on reducing medication-related problems of older adults in the LMICs. Furthermore, future research should look at the integration of pharmacists in the primary care system so that they can provide long-term support for older adults, focusing on the appropriate use of medicines among older adults. It will ease the rising burden of general practitioners in primary care settings and establish pharmacists’ services as an integral element of geriatric-focused primary care service.

## 5. Conclusions

The current review showed that there are several healthcare interventions conducted by community pharmacists for the elderly population. The most common interventions evaluated by RCTs included medication reviews and educational interventions. Moreover, there is evidence in the literature that community pharmacy-based interventions have a beneficial impact on clinical outcomes among older adults, including a reduction in inappropriate medicine use (e.g., sedative–hypnotic drugs), reduction in uncontrolled health problems, and reduction of ADRs. There is limited or inconclusive evidence on the impact of community pharmacists’ interventions on hospitalization, quality of life, and other outcomes from RCTs. Consequently, we believe more research is needed to further investigate the impact of community pharmacists’ intervention on these outcomes. In addition, more structured and tailored interventions are needed to be established at community pharmacies to provide quality services to the elderly population in collaboration with other healthcare professionals (i.e., medical practitioners and nurses) and in an integrated manner within the primary care system.

## Figures and Tables

**Figure 1 healthcare-09-01577-f001:**
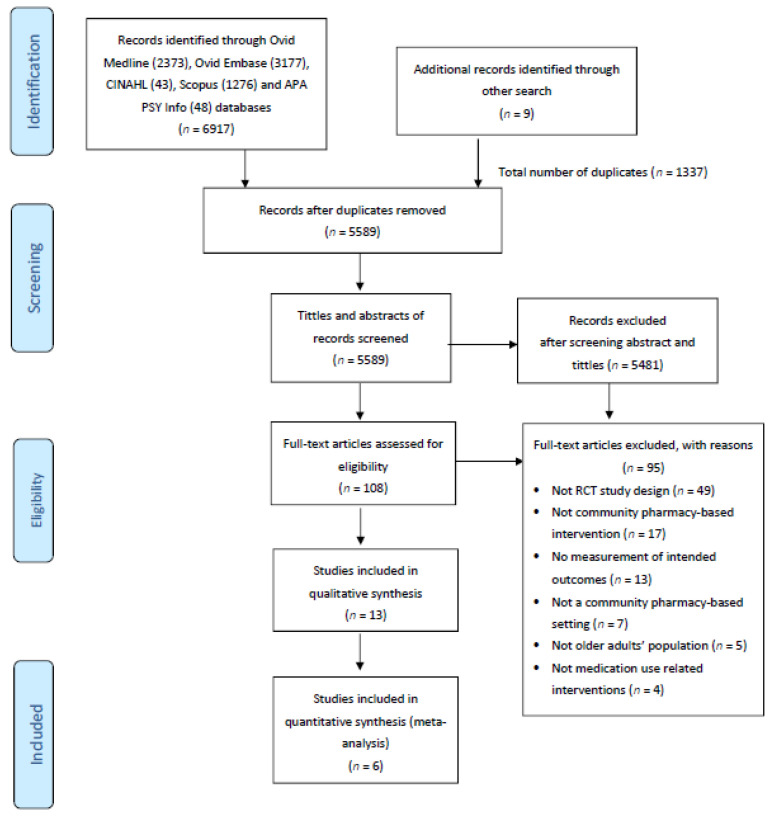
PRISMA flowchart of the selection process.

**Figure 2 healthcare-09-01577-f002:**
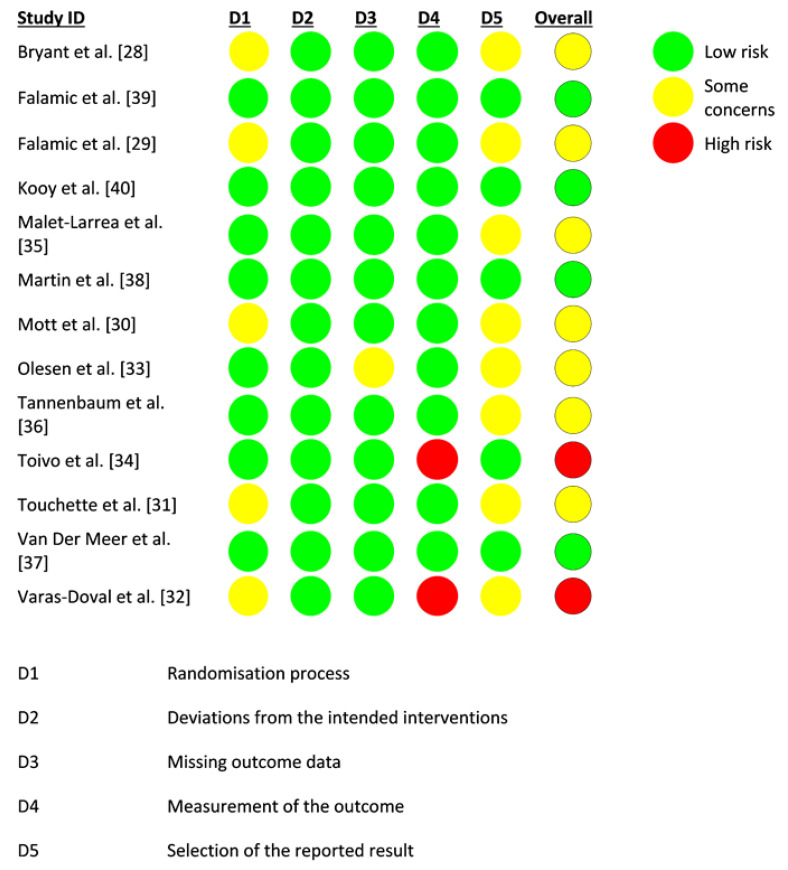
Risk of bias assessment of included studies.

**Figure 3 healthcare-09-01577-f003:**
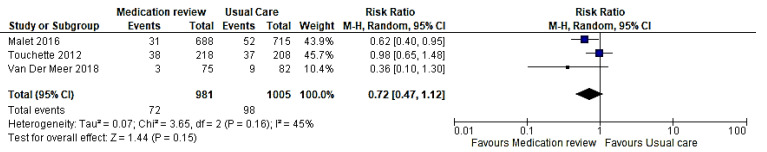
Forest plot showing the risk ratio of older adults hospitalized after medication review.

**Figure 4 healthcare-09-01577-f004:**
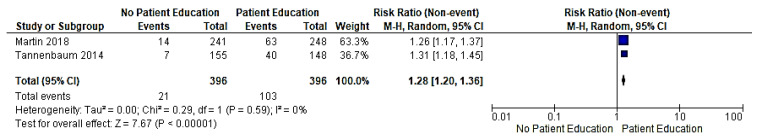
Forest plot showing risk ratio of older adults ceasing benzodiazepine after the patient education intervention.

**Figure 5 healthcare-09-01577-f005:**
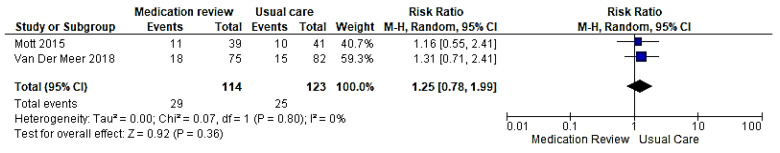
Forest plot showing risk ratio of older adults falls after medication review intervention.

**Figure 6 healthcare-09-01577-f006:**
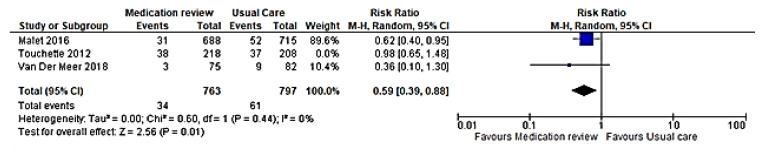
Forest plot of sensitivity analysis showing risk ratio of older adults hospitalized after medication review intervention.

**Figure 7 healthcare-09-01577-f007:**
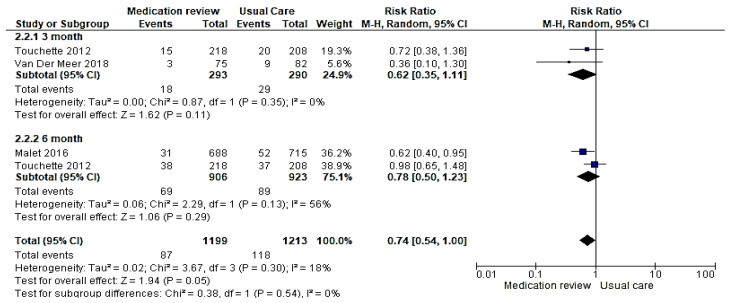
Forest plot showing subgroup analysis of the risk ratio in older adults’ hospitalization according to the duration of follow-up after medication review intervention.

**Table 1 healthcare-09-01577-t001:** Characteristics of included studies.

Author, Year, Country	Study Design, Settings	Interventions	Control Sample Size	Intervention Sample Size	Follow-Up Period	Outcomes	Conclusion
Bryant et al. [28] (2010) New Zealand	Randomized, controlled trial, Community pharmacy	IG: Medication review was completed with the access of medical records from GP. A care plan was prepared, and discussions were completed among CP and GP. Follow-up consultations with patients were completed after taking action on the care plan. CG: Usual care	143	207	6 months and 12 months	Quality of Life (SF-36) and Medication Inappropriateness Index (MAI), number of inappropriate medications	Medication review improved MAI and reduced the number of potentially inappropriate medicines at 6 months follow-up. However, this intervention did not produce a significant improvement in quality of life.
Falamic et al. [29] (2019) Croatia	Randomized, controlled trial, Community pharmacy	IG: Education with follow-up plan (given pillbox and plan form) CG: Standard GP-managed care	66	65	6 months	The incidence and type of adverse drug reactions caused by warfarin	The cumulative incidence of adverse drug reactions was significantly lower in the intervention group.
Mott et al. [30] (2016) United States	Cluster-randomized, controlled trial, Community pharmacy	IG: Medication therapy management with follow-up CG: Received mailed pamphlet describing medication use and falls	41	39	6 months	Rate of discontinuing the fall-risk inducing drug	Medication review significantly improved the rate of discontinuation of fall-risk-inducing drugs among older adults and reduced the number of falls.
Touchette et al. [31] (2012) United States	Randomized, controlled trial, Academic medical center, community pharmacies, and family medicine clinics	IG: Medication therapy management (MTM) with follow-up (enhanced MTM) CG: Usual care	208	Basic MTM = 211 Enhanced MTM = 218	6 months	Frequency of adverse drug events and hospitalization	Medication review did not have a beneficial impact on adverse drug events and hospitalization.
Varas–Doval et al. [32] (2020) Spain	Open-label, multi-center, cluster-randomized, controlled trial, Community pharmacy	IG: Medication review with follow-up CG: Usual care	715	688	6 months	Uncontrolled health problems	Medication review benefited, with a significant reduction in the number of uncontrolled health problems.
Olesen et al. [33] (2014) Denmark	Cluster-randomized, controlled trial, Patient’s home	IG: Pharmaceutical care (examining medication list of older adults, answering any questions on their medications, providing leaflets and motivational adherence support) CG: Usual care	264	253	3, 6, 9, and 24 months	Medication adherence, hospitalization, and mortality	Pharmaceutical care did not bring a beneficial impact on medication adherence, hospitalization, and mortality among older adults.
Toivo et al. [34] (2019) Finland	Cluster-randomized, controlled trial, Community pharmacy, homecare units, public health care center	IG: Collaborative coordination of care (medication review and triage meeting) CG: Standard home care	87	104	12 months	Potentially inappropriate medication	No significant findings were found on the impact of coordination of care on outcomes of older adults’ health.
Malet-Larrea et al. [35] (2016) Spain	Cluster-randomized, controlled trial, Community pharmacy	IG: Medication review with follow-up CG: Usual care	715	688	6 months	Hospitalization	The probability of being hospitalized was 3.7 times higher in the non-intervention group. Thus, medication review had reduced the number of older adults hospitalized.
Tannenbaum et al. [36] (2014) Canada	Cluster-randomized, controlled trial. Community pharmacy	IG: Patient education (materials which also contained benzodiazepine safety and tapering dose) CG: Usual care	155	148	6 months	Benzodiazepine therapy discontinuation	Patient education improved the benzodiazepine discontinuation rate among older adults.
Van Der Meer et al. [37] (2018) Netherlands	Single-blind, randomized, controlled trial, Community pharmacy	IG: Medication review with follow-up CG: Usual care	82	75	3 months	Drug burden index, hospitalization	Medication review did not have significant effects on the number of falls and hospitalization. Moreover, it did not produce an impact on the difference in drug burden index between groups.
Martin et al. [38] (2018) Canada	Cluster-randomized, controlled trial, Community pharmacy	IG: Patient education (education materials were distributed), and education materials were given to prescribers CG: Usual care	241	248	6 months	Sedative-hypnotics (benzodiazepine therapy discontinuation) and potentially inappropriate medication	Patient education reduced the number of benzodiazepine users and reduced the number of inappropriate medications among older adults.
Falamic et al. [39] (2018) Croatia	Prospective, double-blind, randomized, controlled trial, Community pharmacy	IG: Education and follow-up plan with medication review (given a form containing lab values, INR, and pillbox. CG: Usual GP care	66	65	6 months	Time in therapeutic range of warfarin	Patient education improved time in the therapeutic range of warfarin.

**Table 2 healthcare-09-01577-t002:** Subgroup analysis according to the duration of follow-up.

Outcome	Number of Studies	Number of Participants	Statistical Method	Effect Size 95% (CI)
Hospitalization	3 [31,35,37]	1986	Risk ratio (M–H, random, 95% CI)	0.74 (0.54,1.00)
3 months	2 [31,35]	583	Risk ratio (M–H, random, 95% CI)	0.62 (0.35,1.11)
6 months	2 [31,37]	190	Risk ratio (M–H, random, 95% CI)	0.78 (0.50,1.23)

## Data Availability

The authors confirm that data supporting the findings of this study are available within the article and its Appendix A.

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
