# Peer review of "Clinical and Humanistic Outcomes of Community Pharmacy-Based Healthcare Interventions Regarding Medication Use in Older Adults: A Systematic Review and Meta-Analysis"

_healthcare, 2021, doi:10.3390/healthcare9111577_

Round 1

Reviewer 1 Report

Report on manuscript

Clinical and Humanistic Outcomes of Community Pharmacy-Based Healthcare Interventions Regarding Medication Use in Older Adults: A Systematic Review and Meta-Analysis

  • Manuscript ID: healthcare-1406608
  • Journal: Health Care

The current meta-analysis study by Christopher et al., reports the clinical and humanistic outcomes of therapeutic interventions related to medication problems in older adult, at primary care level.

Abstract and Introduction parts are well-structured. There is sufficient connection between literature survey and study design. Methodology described, looks sufficient but can be improved in future study design. Discussion and conclusions are described in a good way.

This study is important as medication non-adherence is one of the major problem in older adults. And community pharmacist can play a pivotal role to counsel patients about proper use of medication. To develop a rational, for chronic patients, about medication is an important factor to have a proper medication adherence and to achieve targeted therapeutic goals. Therefore, it can be beneficial to council patients (by nursing staff and most importantly by clinical and community pharmacist) about the significance of drug use and its mechanism and associated pros & cons to improve their confidence regarding medication use.

Minor

  • Define the abbreviation at its first place e.g., RCTs in Line 19.

Author Response

Point 1: Abstract and Introduction parts are well-structured. There is a sufficient connection between literature survey and study design. Methodology described, looks sufficient but can be improved in future study design. Discussion and conclusions are described in a good way.

Response 1: Thank you very much for your nice words and constructive feedback. We greatly appreciate your support.

Point 2: Define the abbreviation in its first-place e.g., RCTs in Line 19

Response 2: Thank you. We have now defined the abbreviation in its first place. (Page 1, line 19)

Reviewer 2 Report

The authors presented a review that is interesting for practice. However, it needs technical correction.

Line 40 - put a point after, not before square brackets (here and in the text)

Line 359, 380 - Table 1 and Table 2 are need technical correction (please, use the table layout for this journal); it is recommended to modify the contents and column headers of Table 1.

General remarks:

- Please, use the journal template for the font throughout the text and in all tables.

- The Introduction section is large, you can reduce it by leaving the most important items. On the contrary, it is recommended to expand the Discussion section and add a final drawing combining the results obtained by the authors.

Author Response

Point 1: The authors presented a review that is interesting for practice. However, it needs technical correction. Line 40 - put a point after, not before square brackets (here and in the text)

Response 1: Thank you for the positive comments and nice words.  We have now considered your comment for putting a point after not before square brackets in line 40 and throughout the text, (Page 1, line 40 and all throughout text).

Point 2:  Line 359, 380 - Table 1 and Table 2 need technical correction (please, use the table layout for this journal); it is recommended to modify the contents and column headers of Table 1.

Response 2: Thank you very much for the feedback. We have now revised Table 1 and Table 2 to adhere to the journal template layout. In addition, the layout of tables and style of the manuscript are now reviewed by MDPI author services to ensure consistency with the journal style (the certificate is attached).

Point 3: Please, use the journal template for the font throughout the text and in all tables.

Response 3: Thank you. We have reviewed this. In addition, layout, style, and formatting are now are now reviewed by MDPI author services to ensure consistency with the journal style (the certificate is attached).

Point 4: The Introduction section is large, you can reduce it by leaving the most important items. On the contrary, it is recommended to expand the Discussion section and add a final drawing combining the results obtained by the authors.

Response 4: Thank you for the comment. We have now expanded the discussion section by adding a final output of the review based on the current review findings (Page 17, line 498). For the introduction, we believe we need to have this important background to ensure the readers are familiar with the literature and context related to this review and to pave the way for our study. We hope this is now OK.

Reviewer 3 Report

  1. Figure 1 should be on only one page.
  2. Maintain the same style in Table 1.
  3. Figure 2, I suggest maintain only a colors code, the symbols could confuse the readers.
  4. Improve the quality of Figures 3, 4, 5, and 6.
  5. The results section is so descriptive; the authors should improve the form of presenting this section.
  6. The conclusion should be more prepositive and not descriptive. The authors should focus on the relevance of their work.

Author Response

Point 1: Figure 1 should be on only one page.

Response 1: Thank you for the feedback. We have now amended figure 1 to be in one page.

Point 2:  Maintain the same style in Table 1.

Response 2:  Thank you for the comment. We have now revised and maintained the same style in Table 1. In addition, layout, style, and formatting are now are now reviewed by MDPI author services to ensure consistency with the journal style (the certificate is attached).

Point 3:  Figure 2, I suggest maintain only a colors code, the symbols could confuse the readers.

Response 3: Thank you and we appreciate your suggestion. We have now removed the symbols and maintained the colors code only in Figure 2.  

Point 4:  Improve the quality of Figures 3, 4, 5, and 6.

Response 4: Thank you for your feedback. We have now improved the resolution and quality of figures 3, 4, 5, and 6.

Point 5: The results section is so descriptive; the authors should improve the form of presenting this section.

Response 5: Thank you for the constructive feedback. We have further edited this section. Moreover, we needed some description to ensure presenting the interventions and outcomes of the studies at a sufficient level to obtain the differentiation between included studies. This is because there are several interventions and with some different components. Hence, we have tried to describe this to explain the differences in some outcomes. 

Point 6: The conclusion should be more prepositive and not descriptive. The authors should focus on the relevance of their work.

Response 6: Thank you for the comments. We have added some concluding text on focusing our findings and the conclusion drawn in relevance to our review.

Reviewer 4 Report

The topic of this systematic review is very important and current. The manuscript is very well written and with important information on the role of community pharmacists in improving the impact of medicine use in the elderly population.

The aim of the study in the abstract is different from the aim at the end of the introduction. In the abstract, the authors state that the aim is “ to determine the clinical and humanistic outcomes 17 of community-pharmacy-based interventions on medication-related problems of older adults at the 18 primary care level.” However, during the manuscript, it is not evident the analysis of clinical and humanistic outcomes. From reading the results presented, it appears to be a review that analyses outcomes of drug-related problems rather than clinical and humanistic outcomes. So please revise the title of the manuscript.

Lines 128-129 – “Two reviewers (C.M and B.K.C) screened and reviewed the titles and abstracts of 128 identified studies using the search strategy and those from additional sources to identify 129 studies that meet the inclusion criteria as mentioned earlier”. - What other sources are the authors referring to?

In Figure 1 authors identified as the reason for exclusion “not relevant study”, what is considered relevant and what is not considered relevant?  Please give an objective criterion for this classification.

Author Response

Point 1: The topic of this systematic review is very important and current. The manuscript is very well written and with important information on the role of community pharmacists in improving the impact of medicine use in the elderly population.

Response 1: Thank you very much for your nice words. We really appreciate your feedback. 

Point 2: The aim of the study in the abstract is different from the aim at the end of the introduction. In the abstract, the authors state that the aim is “ to determine the clinical and humanistic outcomes of community-pharmacy-based interventions on medication-related problems of older adults at the primary care level.” However, during the manuscript, it is not evident the analysis of clinical and humanistic outcomes. From reading the results presented, it appears to be a review that analyses outcomes of drug-related problems rather than clinical and humanistic outcomes. So please revise the title of the manuscript.

Response 2: Thank you. We have now edited the aims for better clarity and consistency. We added the term clinical and humanistic outcomes at the end of the introduction. We believe the title is appropriate as it determines various outcomes compromising clinical and humanistic outcomes such as hospitalisation, risk of fall, use of hypnotic-sedatives, uncontrolled health conditions, quality of life, adherence etc. So, we prefer to have the current title as it is still concise and describes well the current review various outcomes.)

Point 3: Lines 128-129 – “Two reviewers (C.M and B.K.C) screened and reviewed the titles and abstracts of 128 identified studies using the search strategy and those from additional sources to identify 129 studies that meet the inclusion criteria as mentioned earlier”. - What other sources are the authors referring to?

Response 3: Thank you very much for the feedback. We have now described what the additional sources that we reviewed are in the text.

Point 4:  In Figure 1 authors identified as the reason for exclusion “not relevant study”, what is considered relevant and what is not considered relevant?  Please give an objective criterion for this classification.

Response 4: Thank you for the comment. We now have revised this.

Round 2

Reviewer 3 Report

The figure on page 14 corresponds to Figure 3 on page 11, please eliminate

Author Response

Comment: The figure on page 14 corresponds to Figure 3 on page 11, please eliminate.

Response: It is now done. Thank you very much.

Reviewer 4 Report

In general, the authors responded to all comments and have improved the manuscript. I have only two comments.

I still have some doubts regarding the search methodology and database selection. In the first paragraph of the Results section , the authors wrote: "Another nine articles were retrieved from other sources and websites such as Google Scholar, the Malaysian Pharmaceutical Society website, and from manual search". If authors considered important to search papers at "Google Scholar and the Malaysian Pharmaceutical Society".  Why didn't they do the search initially in all the databases?

Relative to figure 1, what is the difference between the criterion "No outcome measurements" and "Not intended outcomes", Couldn't it be just one criterion? changing the name? 
I also think the figure will be more readable if you change "49 - Not RCT study design" to "Not RCT study design (n=49), in all points.

Author Response

Point 1: In general, the authors responded to all comments and have improved the manuscript. I have only two comments.

Response 1: Thank you for the positive comments and nice words.  We have now considered your two comments.

Point 2:  I still have some doubts regarding the search methodology and database selection. In the first paragraph of the results section, the authors wrote: "Another nine articles were retrieved from other sources and websites such as Google Scholar, the Malaysian Pharmaceutical Society website, and from manual search". If authors considered important to search papers at "Google Scholar and the Malaysian Pharmaceutical Society".  Why didn't they do the search initially in all the databases?

Response 2: Thank you very much for the feedback. In fact, this is a well-established practice to search in standard databases and other sources as stated by PRISMA guidelines. We did the search initially in all the mentioned databases but mention it in the result section. Please see item 7 of the official PRISMA checklist that indicates “Present the full search strategies for all databases, registers and websites, including any filters and limits used.” It can be accessed at: http://www.prisma-statement.org/

Moreover, this is now emphasized in doing systematic reviews. Please refer to the information in Figure 1 in the guidelines published in BMJ in 2021 and can be accessed at: BMJ 2021; 372 doi: https://doi.org/10.1136/bmj.n71

We have also added/amended the following section “Another nine articles were retrieved from other sources such as Google Scholar for grey literature, manual search in the key references retrieved, and other websites particularly the Ministry of Health Malaysia and the Malaysian Pharmaceutical Society website.” (Page 4, para 3)

We hope it is now OK.

Point 3: Relative to figure 1, what is the difference between the criterion "No outcome measurements" and "Not intended outcomes", Couldn't it be just one criterion? changing the name? I also think the figure will be more readable if you change "49 - Not RCT study design" to "Not RCT study design (n=49), in all points.

Response 3: It is now done as per your suggestion. Thank you very much.